# The Use of Mobile Payment Systems in Post-COVID-19 Economic Recovery: Primary Research on an Emerging Market for Experience Goods

**Maiya M. Suyunchaliyeva [1], Raghav Nautiyal [2], Aijaz A. Shaikh [3,*] and Ravishankar Sharma [2,4]** 

1   Management Department, Al-Farabi Kazakh National University, Al-Farabi 71, Almaty 050040, Kazakhstan; suyunchaliyevam@gmail.com
2   Center for Inclusive Digital Enterprise, Wellington 6012, New Zealand; raghavnautiyal26@gmail.com (R.N.); ravishankar.sharma@zu.ac.ae (R.S.)
3   School of Business and Economics, Jyväskylä University, 40014 Jyväskylä, Finland
4   College of Technological Innovation, Zayed University, Abu Dhabi 144534, United Arab Emirates
*   Correspondence: aijaz.a.shaikh@jyu.fi; Tel.: +358-46-9516017

**Abstract:** This study investigated whether mobile payment services could drive post-COVID-19 pandemic recovery in the 'experience goods' sector (e.g., tourism) utilising Bandura's self-efficacy or social cognitive theory. It explored the factors influencing the intention to continue using mobile payment services and the intention to recommend these to others. An empirical survey was conducted to assess the study variables, and the data obtained therefrom were analysed using the industry-standard Cross-Industry Standard Process for Data Mining method. The study results suggest that personal innovativeness and perceived trust influence consumers' intention to continue using mobile payment services and that perceived trust, personal innovativeness and outcome expectancy influence consumers' intention to recommend the use of such services to others. The research findings have filled a research gap in emerging markets and can serve as the basis for formulating a winning marketing and operational strategy for nascent technologies such as mobile payment services. It would be naïve to extract findings from mature markets such as East Asia, the European Union and the United States and to apply these to developing markets. In addition, this study's investigation of the variables that can influence the intention to continue using mobile payment services and to recommend the use of these to others goes into the heart of the sustainability issue because the study's findings can help mobile payment service providers sustain the use of their applications and thus also sustain the advantages as such.

**Keywords:** mobile payments; CRISP-DM method; domestic tourism; COVID-19

## 1. Introduction

Digital technologies have facilitated the access to and convenient use of services (i.e., anytime, anywhere) through various portable and wearable devices, including cell phones and smartwatches. Among these services, mobile payments occupy a crucial position in the financial and payment landscape and in promoting a digital banking culture. For example, consumers need not visit bank branches and can just conveniently and easily download mobile payment applications on their internet-enabled cell phones to conduct various value-added and traditional banking and payment services, such as mobile shopping, fund transfer, utility bill payments, making donations, checking one's bank balance and even locating the nearby ATMs (automated teller machines).

The socioeconomic consequences of the COVID-19 (coronavirus disease 2019) pandemic on the tourism industry to date have been catastrophic [1]. To minimise the damage caused by the pandemic to such industry, several governments in developing and developed countries and several international organisations have implemented a two-pronged

strategy. First, governments and international travel agencies have started promoting domestic or local tourism [2]. Second, non-cash transactions through mobile payment services have been promoted to minimise COVID-19 contamination. Consequently, the tourism and hospitality sectors have witnessed a wider use of contactless mobile payment technologies [3]. This change in the business and payment landscape has provided a much-needed boost to the local tourism industry, which has suffered much from the widespread flight and booking cancellations since the start of the pandemic. The aforementioned 'disruptive' technology has opened new business avenues for banks, other financial institutions and fintech start-ups and has thus prodded them to develop and allow access to cashless mobile payment services using proximity (near-field communication [NFC]) or remote (Net) mechanisms. That is, proximity or contactless payments can be made simply by activating the NFC or Bluetooth option on one's cell phone or smartwatch and placing the cell phone or smartwatch near the point-of-sale terminal, whether using a PIN (personal identification number) or not using one [4,5]. Android Pay and Samsung Pay are famous examples of NFC payment systems. Remote payments can be made from the home or office using these and other mobile payment applications.

Research [6–11] has defined mobile payment applications as any digital payment applications where a mobile device is used to initiate, authorise and confirm an exchange of money with goods and services, and consider these 'star' or 'killer' applications that can increase the financial inclusion in many developing regions. The revolutions in mobile technologies and the availability of low-cost internet/broadband connections have blurred the difference between the developed and developing countries and has reduced the gap between banked and unbanked or remote consumers. Anyone with a cell phone or any portable smart device with internet (Wi-Fi) or GSM (Global System for Mobile communication) connection can access a host of banking and payment services.

Earlier, cell phones were used only for texting and calling. Apple then introduced smartphones in 2007, which disrupted the traditional mobile-technology-based business models and provided new business opportunities. Companies introduced several downloaded mobile applications providing a host of value-added services and cashless payment options. The tourism sector is one of the subsectors of the economy that have benefited the most from these developments. For example, earlier, electronic tickets came to be commonly used in the tourism and leisure sector [12]. The mobile revolution then further modernised the shopping and payment mechanisms and provided better consumer experiences. Moreover, as tourism products and services are perishable, experiential, heterogeneous, and information intensive in nature, they are ideal for digital distribution, such as by using a mobile device [13].

A theoretically sound model based on self-efficacy or the social cognitive theory (SCT) [14] and the trust theory linking the antecedents and outcomes of use continuance intention in the mobile-payment-application context was empirically tested in this study and is presented herein. There was a valid purpose for supplementing SCT with the trust theory. Here, we support the idea [15] that SCT alone cannot explain the intention to continue to use mobile payment applications. This is due to the digital and remote nature of such applications, the distance separating the consumers and the service providers and the absence of human interactions [16]. Thus, trust (including e-trust) in the service provider and in the e-payment application are of paramount importance.

This study was conducted to come up with a conceptual model that predicts mobile payment application usage continuance intention and its relationship with intention to recommend mobile payments as a second key dependent variable. The study participants included domestic travellers and tourists in Kazakhstan who have been using mobile payment applications during their tour. To make the study objectives more explicit, the research questions below were formulated.

RQ1: What are the salient factors determining domestic travellers' mobile payment application usage continuance intention and intention to recommend mobile payments?

RQ2: How do the SCT constructs (outcome expectancy, self-efficacy) and social influence and perceived trust correlate with mobile payment application usage continuance intention?

RQ3: How does mobile payment application usage continuance intention promote or increase the consumer's intention to recommend mobile payments?

Kazakhstan and the domestic travellers and tourists therein were selected as the context of this study for four major reasons. Firstly, Kazakhstan is a highly collectivistic society [17] as it has a strong family system characterised by trust, harmony, and close ties between the family members and thus a strong familial influence. Social events and gatherings are also regularly arranged, and local travels are frequently undertaken. Secondly, the international travel restrictions that have been put in place globally due to the COVID-19 pandemic have promoted and increased the volume of domestic travel and tourism activities during holidays and on weekends in several countries [2,18], including Kazakhstan. Thirdly, such domestic travelers and tourists use mobile payment services frequently [19] as they consider these safer and more convenient than cash transactions while travelling with their family and friends to their dream destinations domestically. Lastly, the phenomenon of the use of mobile payment applications by domestic travellers and tourists coincides with the efforts of the government and other organisations to motivate the citizens to abandon or minimise cash transactions and to instead adopt remote or digital payment systems, including mobile payment systems.

Next, we present the theoretical background of the current study, and in Section 3, the research hypotheses that were formulated for empirical testing are presented. Section 4 describes the field research method that was adopted for this study: The Cross-Industry Standard Process for Data Mining (CRISP-DM) method. The study findings are presented and discussed in Section 5. The paper concludes with a discussion of the key findings, contributions and limitations of the study.

## 2. Theory

### 2.1. The Use of Mobile Payment Services in the Tourism Sector

The hospitality and tourism sector is considered a significant revenue-contributing sector for a country [20]. This sector consists of airline companies, tour operators, hotels, the sharing economy (e.g., Airbnb), car rentals, tourist attractions, shopping malls and restaurants [21]. In the wake of the pandemic, [22] introduced a 'new normal' consisting of new standards and protocols to promote tourism and safe travel. Aside from improved hygiene and social distancing, these included the use of contactless mobile payment systems. It has been widely acknowledged, however, that the slow growth in mobile commerce, including mobile payment services, can harm the hospitality and tourism sector and impede its growth. According to the 2021 Hospitality Industry Trends report released by [23], contactless payment systems and their widespread use are indispensable to the hospitality and tourism industry's survival and growth. This is because of such systems' added efficiency and effectiveness and the useful innovation they introduced in the aforementioned sector [24].

In the aforementioned reports, a strong correlation is found between the hospitality and tourism sector and mobile commerce. Mobile commerce, of which mobile payment is an integral component, is essential to the growth of tourism, and this direct relationship became more stringent after the COVID-19 debacle.

### 2.2. Social Cognitive and Trust Theory

The theoretical model that was developed in this study and is proposed herein is partially rooted in the well-known SCT proposed by [14] and the trust theory proposed by [25]. SCT, with self-efficacy as one of its components, extends the technology acceptance model by providing a more comprehensive understanding of the behavioural intention to adopt technology, system or innovation [26]. In his SCT, Bandura divided the affective and behavioural outcomes into self-efficacy and outcome expectancy. SCT considers self-

efficacy a functional value. It explains how much effort and time consumers are willing to invest in accomplishing a specific task in the face of various obstacles [27].

Since the early 1950s, trust has played a significant role in everyday dealings between consumers or users and companies or service providers. After the advent of e-commerce, m-commerce and other remote or internet-based service and delivery options that blurred or almost eliminated personal interactions, trust appeared as a game changer, a disruptive force. It came to occupy a central position in the development and deployment of new business models. Trust is crucial in digital transactions such as mobile payments, which are traditionally considered highly risky and uncertain. Trust has been extensively examined in the sociology, psychology, economics and management fields [28].

### 2.3. Antecedents of Usage Continuance Intention

### 2.3.1. Self-Efficacy/Personal Innovativeness

According to prior research [29,30], self-efficacy is one of the key drivers of user activity and has direct and indirect impacts on the intention to use and the actual use of different technologies and systems, including mobile payment systems. Self-efficacy (and self-efficacy expectancy) is considered akin to 'personal innovativeness' and 'perceived ease of use' [31] and 'perceived behavioral control' [32]. Ref. [14] defined the term as '[the] belief in one's capabilities to organise and execute the courses of action required to produce given attainments' (p. 3) and as the 'judgments of how well one can execute [the] courses of action required to deal with prospective situations' [27]. Ref. [33] claim that past interactions and experiences and current exposure to a certain technology, system or service contribute to one's self-efficacy, and according to [34], self-efficacy directly or indirectly influences consumers' financial and payment service usage continuance intentions.

There are a few misconceptions about self-efficacy that need clarification for a better understanding of the theory's context. For example, self-efficacy does not consider what people have done in the past concerning the acceptance and usage of a specific technology, service or system. Instead, it considers and makes judgements about what people can do in the future [34,35]. The self-efficacy theory involves people's perception and understanding of how well they can perform a task [36].

### 2.3.2. Outcome Expectancy

A plethora of earlier studies in the fields of physical activity or health research [37] and education research [38,39] have considered using the variable 'outcome expectancy'. The examination of outcome expectancy is rare in the management field. Researchers [40,41] have defined outcome expectancy as an individual's belief that the desired outcome can be attained by accomplishing a task. Ref. [32] have defined the term as the belief that if one engages in a certain behaviour (e.g., contactless payment), a corresponding outcome will follow (e.g., reduction of COVID-19 infections).

The social cognition models have given much attention to the role of self-efficacy and significantly less attention to 'outcome expectancy' [32]. However, self-efficacy has a direct impact on outcome expectancy. The latter is also used as a multi-dimensional variable. In their study on blog sharing, ref. [42] investigated the antecedents of continuous blog sharing. They used outcome expectancy as one of the independent variables in their study and defined it as consisting of three aspects: financial capital, knowledge capital and social capital. Similarly, ref. [43] examined the effects of hedonic and utilitarian outcome expectations on the intention to continue playing online games.

### 2.3.3. Social Influence

Social influence is one of the important constructs used in the unified theory of acceptance and usage of the technology model [44]. In the consumer context, refs. [45,46] defined social influence as the extent to which users or consumers perceive that other important people believe they should use a particular technology, system or service. Social influence is akin to social pressure from subjective norms and is considered an important motivation for adopting and using a new technology or system [47], such as the mobile payment system. Unlike individualist societies mostly found in the Western or developed regions of the world, collectivist societies rely heavily on the suggestions and recommendations of family and friends. Thus, in the latter societies, peer-to-peer communications and social networks play a significant role in people's adoption, continuance use or even abandonment of an application, a service or a system.

### 2.3.4. Perceived Trust (Including Online Trust)

As technologies and information systems evolved and were extensively adopted and used in consumers' everyday lives, virtual connections and interactions became the modus operandi. Physical interactions or interactions requiring face-to-face contact have become near extinct in the developed world, and the developing world is catching up fast in this regard. Trust appears to be one of the most critical aspects of this transition. In other words, developing and keeping a trustworthy relationship with the consumers in the virtual world is of paramount importance. Ref. [48] defined trust in the relevant context as a person's readiness to be open to the actions of another party or service provider. Trust deficiency can severely damage a firm's reputation and can lead to the discontinuation of the service or system. Trust is considered an important factor especially under conditions of uncertainty and risks, and it develops over time through good interpersonal relationships [49].

Online trust or e-trust refers to trust in a digital environment. According to [50], online trust is the phenomenon in which a firm's stakeholders rely on the firm to efficiently and effectively conduct business activities through an electronic medium (its website). Online trust exists in an environment where there is no direct physical contact, where the perceived social and moral pressures are different and where digital devices come into play in place of human interaction [51].

### 2.3.5. Intention to Recommend

The intention to recommend the adoption or use of a service, technology or system is based on several factors, including both commercial and non-commercial ones. For example, according to [52], when choosing a holiday destination, tourists rely on non-commercial sources of information, including the recommendations they receive from their family and friends and from their followers on social media. Therefore, considering its significance in the tourism literature, intention-to-recommend behaviour is an important research area [53]. Intention to recommend is akin to word of mouth (WOM) and is considered a post-adoption behaviour [54], which is in line with the scope of this research, where the continuance intention to use technology was examined.

## 3. Conceptual Model and Research Hypotheses

As shown in Figure 1, the conceptual model that was used in this study included nine hypotheses (outcome expectancy [H1a,b], social influence [H2a,b], personal innovativeness [H3a,b] and perceived trust [H4a,b]) directly related to the consumer's mobile payment technology usage continuance intention and leads to the intention to recommend (H5) mobile payments.

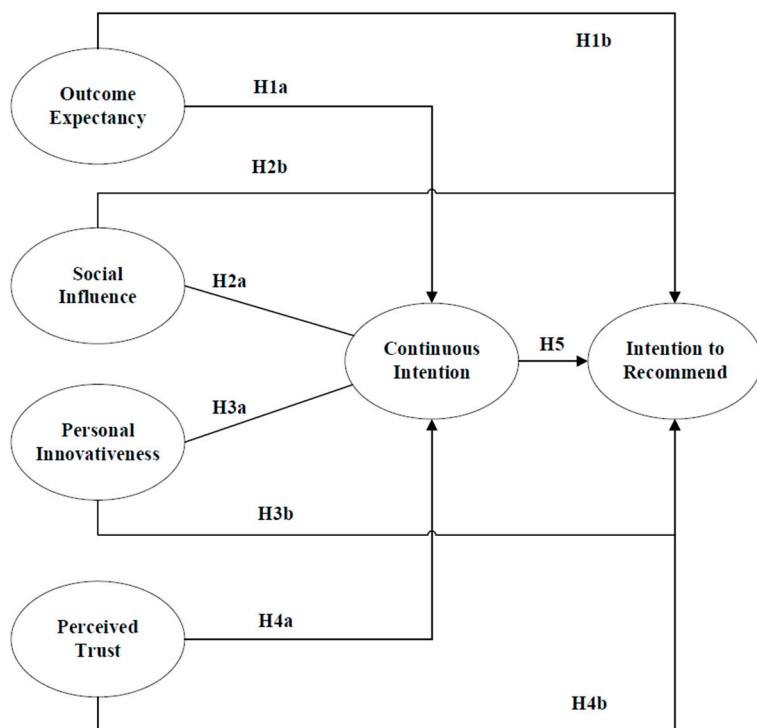

**Figure 1.** Theoretical Model.

*3.1. Relation of Outcome Expectancy to Usage Continuance Intention and Intention to Recommend*

In the first study hypothesis, we intended to examine how outcome expectancy is directly and positively related to mobile payment technology usage continuance intention and intention to recommend mobile payments. This was in line with the past research, where the relationship between outcome expectancy and usage continuance intention was established. For example, [55], while examining the motivational factors influencing consumers' intention to continue using web applications, found that consumers' web application usage continuance intention is directly correlated with their satisfaction with the application and their internet self-efficacy and outcome expectations. Ref. [42] used outcome expectancy as a multi-dimensional construct (financial capital, knowledge capital and social capital). They examined its relationship with blog-sharing continuance intention and found that outcome expectancy for knowledge and social capital encourages usage continuance intention but outcome expectancy for financial capital does not. Ref. [43] examined the relationship between hedonic and utilitarian outcome expectations and their effects on the intention to continue playing online games and found that continuance intention is predicted by utilitarian and hedonic outcome expectations.

Outcome expectancy comes very close to performance expectancy [56]. In a previous study on the acceptance and use predictors of wearable fitness technologies [57], direct relationships were found between performance expectancy, usage continuance intention and intention to recommend. As such relationships were established in the technology context (i.e., wearables, online blogs, online games), they may also be true for mobile payment technologies. Thus, the hypotheses below were formulated.

**Hypothesis 1a (H1a).** *Outcome expectancy is positively related to usage continuance intention.*

**Hypothesis 1b (H1b).** *Outcome expectancy is positively related to intention to recommend.*

### 3.2. Relation of Social Influence to Usage Continuance Intention and Intention to Recommend

The relationship between social influence (similar to subjective norms) and the intention to continue using and to recommend certain products, services or technology is well established in the tourism, information technology and business literature. For example, [58], in their study that aimed to determine the effect of destination image and subjective norms on the intention to visit Lombok Island in Indonesia, found a significant relationship between social influence or subjective norms and tourists' intention to visit. In the context of sustainable rural tourism, ref. [59] found that subjective norm has a significant positive effect on intention to visit rural tourism sites. Ref. [60] examined Chinese consumers' intention to continue using online social networks and found a significant positive relationship between social influence and usage continuance intention. In addition, [61], while examining the factors influencing the intention to continue using Web 2.0, found that social factors have a significant direct impact on usage continuance intention. In the context of mobile payment, it was found in a study [62] that subjective norms, risk, perceived usefulness, customer brand engagement and trust are the most significant antecedents of intention to continue using contactless mobile payment systems. In another study [63], a direct relationship was found between subjective norms and intention to recommend. We thus formulated the hypotheses below.

**Hypothesis 2a (H2a).** *Social influence is positively related to usage continuance intention.*

**Hypothesis 2b (H2b).** *Social influence is positively related to intention to recommend.*

### 3.3. Relation of Personal Innovativeness to Usage Continuance Intention and Intention to Recommend

In recent decades, individual psychological factors such as personal innovativeness have attracted increasing attention in mobile contexts [64]. Ref. [65] found a direct relationship between consumer innovativeness and intention to continue using online check-in services. Ref. [64] also found that personal innovativeness remains an important determinant of usage continuance intention. The relationship between personal innovativeness or self-efficacy and intention to recommend has rarely been examined, but we nonetheless posit that innovative consumers with a predisposition to try out new services, technologies and systems will also be in a better position to recommend or discourage the use of a new product or service, including mobile payment services. Thus, we formulated the hypotheses below.

**Hypothesis 3a (H3a).** *Personal innovativeness is positively related to usage continuance intention.*

**Hypothesis 3b (H3b).** *Personal innovativeness is positively related to intention to recommend.*

### 3.4. Relation of Perceived Trust to Usage Continuance Intention and Intention to Recommend

Reference [66] found a direct relationship between perceived trust and the intention to continue using social networking sites such as Facebook. A similar direct and significant relationship between perceived trust and usage continuance intention was also found by [67] in the context of web-based online banking services. Reference [68] found that customer trust is positively associated with consumers' intention to continue using mobile payment systems in China. Moreover, the correlation between perceived trust and intention to recommend or positive WOM can also be found in the literature. For example, ref. [69], while examining the effects of perceived justice on recovery satisfaction, trust, WOM and revisit intention in an upscale hotel, found that trust is positively associated with WOM and revisit intention. Given these significant findings, we formulated the hypotheses shown below.

**Hypothesis 4a (H4a).** *Perceived trust is positively related to usage continuance intention.*

**Hypothesis 4b (H4b).** *Perceived trust is positively related to intention to recommend.*

*3.5. Relation of Usage Continuance Intention to Intention to Recommend*

The early adopters of any new technology, system or service can influence the success or failure of the technology, system or service by making recommendations in relation to such. Consequently, usage continuance intention is influenced by intention to recommend. Earlier findings have endorsed and established this direct relationship between the two variables. For example, ref. [70] found a direct relationship between the intention to continue using smart fitness wearables and the intention to recommend their use. Similar findings were reported by [57] in the context of social networking platforms. We thus came up with the hypothesis below.

**Hypothesis 5 (H5).** *Usage continuance intention has a direct and positive influence on intention to recommend.*

## 4. Field Research Method

The CRISP-DM method was used for the empirical analysis of the adoption of the mobile payment technology in an emerging market (See Figure 2).

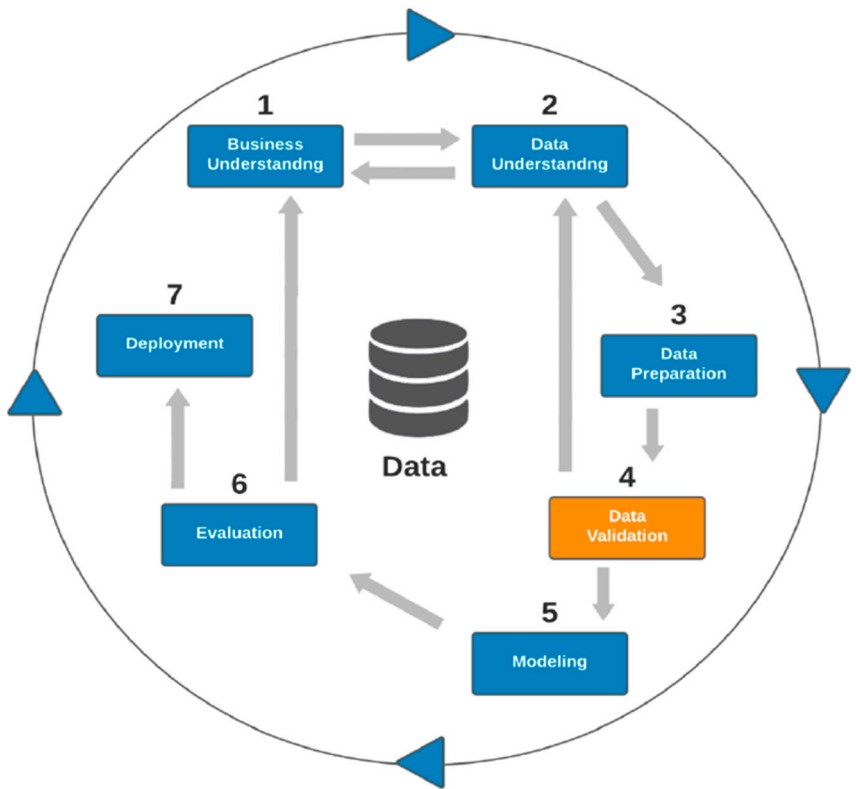

**Figure 2.** Cross-Industry Standard Process for Data Mining (CRISP-DM) Method.

CRISP-DM is preferred by industry particularly because its iterative steps are well-suited for deep and exploratory investigation of business problems. As Big Data Architect Anshul Roy states (https://www.linkedin.com/pulse/chapter-1-introduction-crisp-dm-framework-data-science-anshul-roy/, accessed on 14 September 2021): *The CRISP-DM methodology provides a structured approach to planning a data mining project. It is a robust and well-proven methodology.* The procedures that were used in this study for data collection and pre-processing are briefly described below.

### 4.1. Empirical Survey, Sampling, and Data Collection

A questionnaire was developed to assess the respondents' behavioural attitudes toward the variables relevant to the current study. A list of questions was created through an extensive literature review to assess each composite factor (outcome expectancy, social influence, personal innovativeness, perceived trust, usage continuance intention and intention to recommend), resulting in a 21-item questionnaire with a Likert-scale-type response scheme that captures behavioral intention.

The survey was pre-tested on a group of university students in Almaty, Kazakhstan. The survey was modified on the basis of the feedback received from the pilot test participants. The web-based Webropol application was used to construct the final online survey instrument and to collect data through it. Webropol provides the most versatile, scalable and secure platform for developing and conducting online surveys. One of the study co-authors came from Kazakhstan and understood the local culture and network therein. The link to the survey was shared with university students, faculty and staff members and with others outside universities. A purposeful sampling technique was used to recruit the respondents. The inclusion criteria were as follows: had used mobile payment applications for the last 6 months or more, had gone on local or domestic tours in Kazakhstan one or more times during the COVID-19 pandemic and had used mobile payment applications during such tours.

Appendix A contains the survey instrument that was used for data collection. The data were collected from May to September 2020, when COVID-19 was officially declared a pandemic and was already shattering the world economy, including the tourism sector. At that time, the people had already started to support local or domestic tourism, which provided sufficient motivation and a valid ground for this study.

There were four classifiers and two targets. To assess the inter-item reliability, the Cronbach's alpha coefficient was calculated for the items measuring each variable. All the Cronbach's alpha coefficients were in the range of 0.85–0.93, indicating the high internal consistency of the questionnaire. The descriptive statistics and the demographic details of the respondents are shown in Table 1. They do not indicate a biased or skewed sample.

### 4.2. Data Pre-Processing

The dataset had a few missing values. To address these, the dataset was sent back to its original transcriber, who supplied the missing values. The dataset also had a few values that were considered improbable. These were replaced by computing for the mean of all the other responses to the question. Two sub-datasets were created from the available data. The first dataset contained the itemised scores based on each item in the survey questionnaire. The second dataset contained the composite score for each question based on the items asking about it in the survey questionnaire. As the data were in seven-point Likert scale format, there was no need to encode them. In addition, data normalisation was not required because all the values were on the same scale. For ease of reference, Online Annex B contains a Data Dictionary of the Dependent, Independent and Composite Variables used in the data analytics.

**Table 1.** Demographic profile of the respondents.

| Demographic Categories | Frequency | Percentage (%) |
|---|---|---|
| Gender | 400 | 100 |
|     Male | 136 | 34 |
|     Female | 264 | 66 |
| Age group | 400 | 100 |
|     $\leq$18 years | 79 | 19.75 |
|     19–24 years | 182 | 45.5 |
|     25–34 years | 36 | 9 |
|     35–44 years | 52 | 13 |
|     45–54 years | 31 | 7.75 |
|     $\geq$55 years | 20 | 5 |
| Experience | 400 | 100 |
|     01–03 months | 96 | 24 |
|     04–06 months | 116 | 29 |
|     07–12 months | 89 | 22 |
|     13–24 months | 99 | 25 |
|     $\geq$25 months | 00 | 00 |
| Frequency | 400 | 100 |
|     01–03 times | 63 | 16 |
|     04–06 times | 31 | 8 |
|     07–12 times | 121 | 30 |
|     13–24 times | 96 | 24 |
|     $\geq$25 times | 89 | 22 |
| Profession | 400 | 100 |
|     Student | 232 | 58 |
|     Employee/professional | 50 | 12.5 |
|     Entrepreneur (self-employed) | 65 | 16.25 |
|     Retired | 31 | 7.75 |
|     Unemployed | 19 | 4.75 |
|     Out-of-bound values | 3 | 0.75 |
| Education | 400 | 100 |
|     High school | 39 | 9.75 |
|     Bachelor | 255 | 63.75 |
|     Master | 74 | 18.5 |
|     Ph.D. | 32 | 8 |
| Annual income (tenge) | 400 | 100 |
|     Less than 200,000 | 247 | 61.75 |
|     200,001–400,000 | 84 | 21 |
|     400,001–600,000 | 42 | 10.5 |
|     600,001–800,000 | 25 | 6.25 |
|     More than 800,001 | 2 | 0.5 |

*4.3. Computing Correlations (Composite Independent Variables)*

The pandas Python library was used to compute Pearson's correlation coefficients among the composite independent variables (See Table 2).

*4.4. Generating a Correlation Matrix for the Itemised Values of Intention to Recommend and Usage Continuance Intention*

The pandas Python library was next used to find the correlations between the itemised values of intention to recommend (IR) and usage continuance intention (CI). Table 3 shows that most of the correlations are strong and suggests consistency among and between IR and CI.

**Table 2.** Correlation Coefficients of Composite Independent Variables.

| Variables | (1) | (2) | (3) | (4) |
|---|---|---|---|---|
| (1) Outcome expectancy | 1 | 0.71 | 0.70 | 0.73 |
| (2) Social influence | | 1 | 0.74 | 0.68 |
| (3) Personal innovativeness | | | 1 | 0.73 |
| (4) Perceived trust | | | | 1 |

\* $R^2$: $-0.1379$

**Table 3.** Correlation Coefficients of the Itemized Dependent Variables.

| Variables | (1) | (2) | (3) | (4) | (5) |
|---|---|---|---|---|---|
| (1) CI future | 1 | 0.73 | 0.68 | 0.64 | 0.58 |
| (2) CI daily | | 1 | 0.67 | 0.53 | 0.55 |
| (3) CI frequency | | | 1 | 0.69 | 0.66 |
| (4) IR recommendation | | | | 1 | 0.86 |
| (5) IR subscribe | | | | | 1 |

*4.5. Conducting Regression Analysis*

The stats models Python library was used to conduct ordinary least squares (OLS) regression analysis on the dataset. Occupation, age group and duration of use were first used as control variables for the regression analysis. Data on occupation were taken from the dataset, and occupation was categorised as either employed or unemployed. Age group was categorised as less than 24 years, 24–44 years or above 44 years. Lastly, usage was classified as less than 6 months, 6–24 months or more than 24 months. OLS regression analysis was conducted to determine if there are correlations between the independent variables and the target. To test for potential endogeneity, the Wu-Hausman test was also conducted during the regression analysis. For this, the linear models Python library was used. The Wu-Hausman test was conducted to determine if there are correlations between the independent variables and the error terms. Occupation, age group and duration of use were also used for this test to maintain consistency (See Table 4).

**Table 4.** Results of linear regression analysis.

| Independent Variables | β | ρ |
|---|---|---|
| Outcome expectancy | 0.1626 | 0.081 |
| Social influence | 0.0724 | 0.937 |
| Personal innovativeness | 0.4460 *** | 0.000 |
| Perceived trust | 0.4396 *** | 0.000 |
| Occupation | 0.0808 | 0.488 |
| Age group | −0.0337 | 0.557 |
| Duration of use | 0.0507 | 0.517 |
| $R^2$ | 0.760 | |
| Adjusted $R^2$ | 0.755 | |
| F-value | 176.9 | |

*** $\rho < 0.001$, ** $\rho < 0.01$, * $\rho < 0.05$.

On the basis of the regression analysis results, the itemised values for the important variables (personal innovativeness, and perceived trust for continuance intention; Outcome expectancy, personal innovativeness and perceived trust for intention to recommend) were chosen as they were deemed useful for prediction. The significant variables were those whose *p*-values were less than 0.05 in the regression analysis. Towards this end, three

machine learning models were created for each independent variable. These shall be discussed further in Section 5.

### 4.6. Splitting the Dataset

The dataset was split evenly into training and test values, with 70% of the values being for training and 30% for testing the accuracy of the model. The pandas Python library was then used to load the dataset and to manipulate it whereas the scikit-learn Python library was used to conduct the experiments on the previously described data (See Tables 5 and 6). Three ML algorithms used were: (i) Tree-based model: Random forest; (ii) Bayesian-network-based model: Naive Bayes; and (iii) Neural-network-based model: Multilayer perceptron (MLP)

(a) *Applying three different machine learning models to predict usage continuance intention*

**Table 5.** Comparison of ML Algorithms for the Prediction of Usage Continuance Intention.

| Independent Variables | Random Forest | Bayesian Network | Neural Network |
|---|---|---|---|
| Correctly classified instances | 101 | 100 | 97 |
| Percentage of correctly classified instances | 84.16 | 83.33 | 80.83 |
| Incorrectly classified instances | 19 | 20 | 23 |
| Percentage of incorrectly classified instances | 15.84 | 16.67 | 19.17 |
| Total no. of instances | 120 | 120 | 120 |

(b) *Applying three different machine learning models to predict intention to recommend*

**Table 6.** Comparison of ML Algorithms for the Prediction of Intention to Recommend.

| Independent Variables | Random Forest | Bayesian Network | Neural Network |
|---|---|---|---|
| Correctly classified instances | 101 | 104 | 101 |
| Percentage of correctly classified instances | 84.16 | 86.67 | 84.16 |
| Incorrectly classified instances | 19 | 16 | 19 |
| Percentage of incorrectly classified instances | 15.84 | 13.33 | 15.84 |
| Total no. of instances | 120 | 120 | 120 |

### 5. Findings

The nature of our business challenge was to understand and then predict if respondents with a given profile would be more likely to continue to use mobile payment services (CI, continuance intention), and if so, if they would recommend the use of mobile payment services to others (IR, intention to recommend). Fundamentally, this is a 'yes-no-maybe' classification problem. In other words, when given user characteristics such as the dependent variables used in the survey, the classifier needs to predict if a given user will continue using mobile payment services and/or recommend their use to others.

A description of ML classifiers is beyond the scope of this paper. Mohammad Waseem (https://www.edureka.co/blog/classification-in-machine-learning/, accessed on 14 September 2021) provides a practical tutorial on this subject in a recent blog of his. While industry professionals routinely use several ML algorithms in the training phase and select the best-performing one among them for prediction purposes, our purpose here was only to show a consistent trend across the ML models. Among the many ML algorithms for classification, we used the random forest, naïve Bayesian and artificial neural network approaches to provide a broad basis for comparison. The strengths of all these algorithms are that they are accurate/stable, fast and tolerant. Tables 5 and 6 suggest that for the prediction (classification) of both intention to recommend (IR) and intention to continue use (CI), all the three aforementioned ML techniques showed above 80% accuracy. The percentage of incorrectly classified instances ranged from 15 to 20%. This is a generally acceptable level of performance for ML applications. However, upon probing further (cf. Online Annex C.5 for Output from Python Scikit), we found that generally speaking,

all three ML algorithms performed better (in terms of recall and precision) for the 'yes' classifications than for the 'no' and 'maybe' classifications. In other words, ML better predicts intention to recommend and continue use than the lack of such intention. This is confirmed by the F-statistic for the latter, which is less than 0.5, and in some instances, 0.

Our data analytics also explored the factors that influence consumers' intention to continue using mobile payment services and to recommend the use of these to others. Firstly, the study found that the random forest algorithm (with 84.16% accuracy) is the best for predicting the intention to continue using mobile payment services whereas the naïve Bayesian algorithm (with 86.67% accuracy) is the best for predicting the intention to recommend mobile payment services. Not too much should be read into this, however, as the performance increase is not statistically significant. The point being made here is that ML is suited for the prediction of the intention to continue using mobile payment services and to recommend the use of these to others when profiles are constructed using some of the independent variables shown in Figure 3, such as personal innovativeness and perceived trust. However, outcome expectancy and social influence are not significant contributors to or predictors of such intentions.

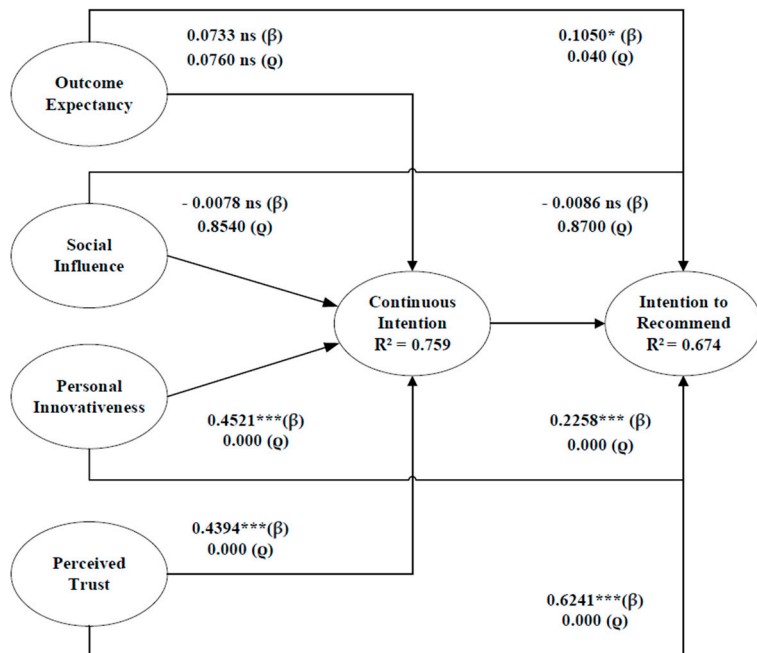

**Figure 3.** Model Derived from Linear Regression Analysis.

Secondly, the regression analysis showed that perceived trust and personal innovativeness are highly correlated with the intention to continue using mobile payment services whereas social influence and outcome expectancy are not significantly correlated with such. For the intention to recommend the use of mobile payment services to others, the regression analysis showed that perceived trust, personal innovativeness and outcome expectancy are highly correlated with it whereas social influence is insignificantly correlated with it. Finally, the relationship between use continuance intention and intention to recommend was found to be significant (see Figure 3).

## 6. Discussion and Conclusions

The COVID-19 pandemic and the associated social distancing, quarantine and isolation protocols put many aspiring tourists in a state of limbo, unable to explore their dream destination since early 2020. The alternatives have been explored, and domestic tourism including the staycation and holistay appeared to be the best alternative. This study examined the experiences of those who had taken local or domestic tours during the pandemic and their experience of using mobile payment applications as their ultimate payment tools. The study thus explored the factors that influence the intention to continue using the mobile payment technology during domestic tour and the intention to recommend mobile payments to others. Several implications can be drawn in light of the findings obtained from the empirical data gathered in the study.

### 6.1. Theoretical Implications

The regression analysis shows that perceived trust and personal innovativeness are highly correlated with the intention to continue using mobile payment services. This coincides with the earlier findings reported by [65–67]. Secondly, social influence and outcome expectancy were found not to be significantly correlated with usage continuance intention. This contrasts with the earlier finding reported by [42,55] that there are direct relationships between social influence, outcome expectancy and usage continuance intention. For intention to recommend mobile payment services, the regression analysis showed that perceived trust, personal innovativeness and outcome expectancy are highly correlated with intention to recommend whereas social influence is not significantly correlated with it. The study also found that the random forest algorithm (with 84.16% accuracy) is the best for predicting the intention to continue using mobile payment services whereas the naive Bayes algorithm (with 86.67% accuracy) is the best for predicting the intention to recommend mobile payment services. Nonetheless, a direct and significant relationship was found between usage continuance intention and intention to recommend, which coincides with the earlier finding reported by [70]. The data visualisation revealed this relationship between the usage continuance and recommendation intentions.

### 6.2. Managerial Implications

This study's findings have several managerial implications that can help the industry in Kazakhstan and beyond formulate a winning marketing and operational strategy for mobile payment services. First, mobile payments have become an integral part of consumer life. For this, much credit goes to the pandemic situation and the emergence of the 'digital natives' (i.e., Gen Z and Alpha), who prefer accessing remote services using innovative mobile payment applications to visiting brick-and-mortar stores. The industry should thus give more attention to fulfilling the growing needs and demands of the digital natives, who are considered innovative by nature and eager to try out new things.

Moreover, perceived trust and personal innovativeness appeared to be the most significant variables in this study driving users' intention to continue using mobile payment applications. Perhaps due to the remote nature of mobile payments, the lack of face-to-face interaction and the nature of the financial transactions, companies must develop and maintain consumer trust. New regulatory frameworks such as the revised Payment Services Directive (PSD2) and General Data Protection Regulation (GDPR) implemented in Europe, and their replication across the globe, have provided much support and comfort to consumers; the same should be ensured when deploying mobile payment services. After all, consumer trust in the service and service provider is of paramount importance for mobile payment applications.

Although unlike other studies conducted on the same topic, this study did not find correlations between social influence, outcome expectancy and usage continuance intention, the industry should not overlook these important elements, especially social influence. Many still seek the opinions of their family and friends, and their decisions are influenced by what others say, recommend and believe.

The growing digital divide between the rural (or remote) and urban households in most developing countries is a greater concern for industry, successive governments, and policymakers [71]. The lack of digital literacy causes such digital divide [72], which inhibits technological advancements, including the adoption and use of mobile payment systems and associated services by a wider segment of the population. With that in mind, the industry, in collaboration with the policymakers and regulators, should adopt a two-pronged strategy. Firstly, the industry should improve the digital literacy of the less privileged segment of society to promote inclusive development and the wider and more frequent use of mobile payment systems and applications. Secondly, the hospitality and tourism sector should develop super-mobile applications equipped with several modules, such as those supporting m-shopping, m-bookings and voice assistance.

### 6.3. Limitations and Future Research Directions

Among the major limitations of this study was the use of purposeful sampling, which might have decreased the degree of representativeness of the study sample. The second major limitation was that the cross-sectional study was conducted at a specific time and place, making it impossible to establish a true cause–effect relationship. A longitudinal study involving a longer time duration and a bigger sample size is thus recommended because mobile payment services are emerging and novel services and the findings on them from a longitudinal study can reveal various related phenomena [71,72].

It is suggested that the future research strive to expand the scope of the services offered via mobile payment applications and examine the use of artificial intelligence (AI) tools and applications in mobile payment applications and how these AI-based mobile applications navigate consumer behaviour, choices and continuance application usage. Also, the theoretical model developed and used in this study can be replicated and used in another context. The implementation of various regulations and standards, such as GDPR and PSD2, has revolutionised the financial-services landscape; the same should be examined to understand the level of and changes in consumer trust in these services and the consumers' level of awareness of these services.

Another important recommendation is that cross-country studies be conducted within the mobile-payments field, such as comparing a Western country with a non-Western country. Opinions regarding the adoption and use of mobile payment services and applications can also be compared by gender. As consumer trust in remote payment services is crucial, it thus requires attention. Future studies can examine the trust issue from the perspective of services and institutions, such as how trust in a mobile payment service provider or institution is formed and how trust in the mobile payment service provider or institution operates under general regulatory conditions. Therefore, micro-level (institution) and macro-level (regulator) trust should be examined in the context of mobile payments.

**Author Contributions:** Conceptualization, M.M.S. and A.A.S.; Data collection, M.M.S. and A.A.S.; methodology, R.N. and R.S.; data analysis, R.N.; writing—original draft preparation, M.M.S. and A.A.S.; writing—review and editing, all the authors; supervision, A.A.S. All authors have read and agreed to the published version of the manuscript.

**Funding:** This research received no external funding.

**Institutional Review Board Statement:** Not applicable.

**Informed Consent Statement:** Not applicable.

**Data Availability Statement:** Not applicable.

**Conflicts of Interest:** The authors declare no conflict of interest.

# Appendix A

**Table A1.** Constructs and Indicators.

| Construct | Indicators |
|---|---|
| Outcome expectancy | I think that using a mobile payment app will enable me to accomplish certain tasks more quickly during a tour.<br>I think that using a mobile payment app during a tour will increase my productivity.<br>If I use a mobile payment app during a tour, it will increase my output for the same amount of effort. |
| Social influence | The people who are important to me think that I should use a mobile payment app during domestic tours.<br>The people who influence my behaviour think that I should use a mobile payment app during domestic tours.<br>My family/relatives have influenced my decision to use a mobile payment app especially during domestic tours.<br>People who are important to me recommend that I use a mobile payment app during domestic tours.<br>People who are important to me view the use of mobile payment apps as beneficial.<br>People who are important to me think that it is a good idea for me to use a mobile payment app during tours. |
| Self-efficacy or personal innovativeness | If I hear about a new mobile payment app, I will look for ways to experiment with it.<br>Among my peers, I am usually the first to explore a new mobile payment app on my smartphone and/or tablet.<br>I like to experiment with using new mobile payment apps for financial services.<br>In general, I am hesitant to try out new mobile payment apps for financial services. |
| Mobile payment application use continuance intention | I intend to continue using mobile payment apps in the future.<br>I will always try to use mobile payment apps in my daily life.<br>I plan to continue using mobile payment apps frequently. |
| Perceived trust | Mobile payment apps can competently and efficiently handle my financial transactions.<br>I believe that my use of a mobile payment app will be in my best interest.<br>I believe that mobile payment apps can be trusted at all times. |
| Intention to recommend | I would like to recommend to others that they subscribe to mobile payment services.<br>If I have a good experience with a mobile payment app, I will recommend to my family and friends that they subscribe to the service.<br>I will recommend to my family and friends that they subscribe to an available mobile payment service. |

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
