# Peer review of "The Use of Mobile Payment Systems in Post-COVID-19 Economic Recovery: Primary Research on an Emerging Market for Experience Goods"

_sustainability, doi:10.3390/su132413511_

Round 1
Reviewer 1 Report
The study offers an interesting analysis of the use of mobile payment systems in the post-covid tourism recovery context. However, there are some theorizing and methodological issues that the authors need to be aware of. Let me offer more details of my comments below.
The introduction is well depicted and the three RQs are clearly theoretically supported. However, the proposed overview is missing a clear focus on the payment system in the tourism sector. I strongly suggest the author turn the spotlight on the payment system in the context of experience goods and in particular on the tourism sector since its introduction. In particular, it is necessary to highlight why these RQs are relevant in the tourism context and more specifically on Kazan tourism?
The literature review, as well as the research hypotheses, are well developed. I just suggest the authors to extend the theory by discussing previous studies investigating these variables within the tourism literature.
How data have been collected? When?
79 respondents are younger than 18 years. How old are they? What are the minimum and maximum ages of respondents? Please report this data when discussing demographics.
I suggest the authors to include a table with items, original references, loadings, p-values, alphas for each construct. Further a single correlation matrix should be provided instead of Tables 2 and 3.
Regression analysis: Age groups are not balanced. It would be better to leave the age variable as collected originally. While it would be beneficial to report the sample composition by the duration of use (experience) of mobile payment in table 1.
Please provide a critical analysis of the results derived by the prediction models defined by the machine learning approach.
Findings are quite tricky to understand. They first discuss results of the machine learning method, then discuss results derived by the regression and finally provides results of a structural model not developed in the method section. How the previous two methods may generate a SEM? Please provide internal and external validity of constructs as well as model fit indexes and a description of the SEM technique implemented.
Theoretical implications should be discussed in light of previous findings in the tourism literature. While some more effort should be made to propose actionable managerial implications.
I hope the comments above are useful for the authors. Good luck!
Author Response
Comment: The study offers an interesting analysis of the use of mobile payment systems in the post-covid tourism recovery context. However, there are some theorizing and methodological issues that the authors need to be aware of. Let me offer more details of my comments below.
Response: Thank you for allowing us to revise and resubmit the manuscript. Point-by-point responses to your comments are discussed below.
Comment: The introduction is well depicted and the three RQs are clearly theoretically supported. However, the proposed overview is missing a clear focus on the payment system in the tourism sector. I strongly suggest the author turn the spotlight on the payment system in the context of experience goods and in particular on the tourism sector since its introduction.
Response: Thank you for identifying this. We have added two paragraphs in the intro section discussing the significance of using the payment systems in the tourism sector especially in the context of local or domestic tourism. In addition, a separate sub-section on “mobile payments in the tourism sector” has been added in the theory section.
Comment: It is necessary to highlight why these RQs are relevant in the tourism context and more specifically on Kazan tourism?
Response: In the paragraph succeeding the research questions, we have explained the rationale for considering the domestic travellers and tourists in the Kazakhstan context and provided three major reasons. Nonetheless, considering your comment, we have refined the RQs as well as the succeeding paragraph for better understanding.
Comment: The literature review, as well as the research hypotheses, are well developed. I just suggest the authors extend the theory by discussing previous studies investigating these variables within the tourism literature.
Response: While carefully addressing your comment, we have added one new sub-section in the theory section entitled “The Usage of Mobile Payments in the Tourism Sector”.
In addition, we have revisited the hypothesis section and found that some of the hypothesised relationships are properly justified and explained in light of the prior literature and in the context of hospitality & tourism, and mobile payments. Thus, some of our hypotheses have a supporting but relevant theoretical background. Nonetheless, we revised the rest of the hypothesis with the tourism literature where available. Please see below the newly added citations in section 3.
- Ramadhani, S. A., Kurniawati, M., & Nata, J. H. (2020). Effect of Destination Image and Subjective Norm toward Intention to Visit the World Best Halal Tourism Destination of Lombok Island in Indonesia. KnE Social Sciences, 83-95.
- Joo, Y., Seok, H., & Nam, Y. (2020). The moderating effect of social media use on sustainable rural tourism: A theory of planned behavior model. Sustainability, 12(10), 4095.
Nonetheless, where the relevant studies conducted in the tourism context were not available, we extend the search criteria, and the variables examined in the context of mobile technology, smart devices, wearable fitness technology, and social media were considered and included in order to justify the hypothesised relationships. Please check the following literature/references:
- Hsu, M. H.; Chiu, C. M.; Ju, T. L. Determinants of continued use of the WWW: an integration of two theoretical models. In-dustrial Management & Data Systems. 2004, 104, 766–775.
- Talukder, M. S.; Chiong, R.; Bao, Y.; Malik, B. H. Acceptance and use predictors of fitness wearable technology and intention to recommend: An empirical study. Industrial Management & Data Systems. 2019, 119, 170–188.
- Chen, S. C.; Yen, D. C.; Hwang, M. I. Factors influencing the continuance intention to the usage of Web 2.0: An empirical study. Computers in Human Behavior. 2012, 28, 933–941.
- Liébana-Cabanillas, F.; Singh, N.; Kalinic, Z.; Carvajal-Trujillo, E. Examining the determinants of continuance intention to use and the moderating effect of the gender and age of users of NFC mobile payments: a multi-analytical approach. Information Technology and Management. 2021, 22, 1–29.
- Melnikov, S.; Aboav, A.; Shalom, E.; Phriedman, S.; Khalaila, K. The effect of attitudes, subjective norms and stigma on health‐care providers' intention to recommend medicinal cannabis to patients. International Journal of Nursing Practice. 2021, 27, 1–10.
Comment: How data have been collected? When?
Response: We apologize for this omission. The details concerning the survey data collection including the timelines have been added and explained in sub-section 4.1, which is now renamed as “4.1 Empirical Survey, Sampling, and Data Collection”.
Comment: 79 respondents are younger than 18 years. How old are they? What are the minimum and maximum ages of respondents? Please report this data when discussing demographics.
Response: Instead of relying on the age groups, the focus was placed on those respondents which had fulfilled the sample criteria used in this study. For example, the purposeful sampling technique was used to recruit the respondents. The sample criteria were based on three conditions. First, the respondents should have used mobile payment applications for the last six months or more. Second, the respondent had undertaken local or domestic tours in Kazakhstan once or more during the COVID. Third, he or she should have used the mobile payment applications during the tour.
The respondents who fall within the age group of 18 or below include those who had fulfilled all these three conditions. For ethical and legal reasons, we did not approach any minor below 16 to participate.
Comment: I suggest the authors include a table with items, original references, loadings, p-values, alphas for each construct. Further a single correlation matrix should be provided instead of Tables 2 and 3.
Response: Since Table 2 was correlations among Composite Independent Variables while Table 3 was among Itemised Dependent Variables, we could not merge them. However, we have provided explanatory notes in the text of Section 4.4.
About the survey items, Annexure A has been developed and added in the revised manuscript containing the description of the items used in this study.
Comment: Regression analysis: Age groups are not balanced. It would be better to leave the age variable as collected originally. While it would be beneficial to report the sample composition by the duration of use (experience) of mobile payment in table 1.
Response: We have omitted age groups from the analysis precisely for the reason given in the comment – they are not balanced.
Furthermore, we provide descriptive statistics of “experience” and “frequency of use” with mobile payment in Table 1 “Demographic profile of the respondents”.
Comment: Please provide critical analysis of the results derived by the prediction models defined by the machine learning approach.
Response: Thank you for identifying this. We have now added the required analysis by way of rephrasing and expanding Section 5 “Findings”.
Comment: Findings are quite tricky to understand. They first discuss results of the machine learning method, then discuss results derived by the regression, and finally provides results of a structural model not developed in the method section. How the previous two methods may generate an SEM? Please provide internal and external validity of constructs as well as model fit indexes and a description of the SEM technique implemented.
Response: We did not use SEM and have re-labeled Figure 3 as – Model Derived from Linear Regression Analysis. Next, we applied 3 simple ML algorithms to ascertain whether the decision of Intention to Recommend and Continuance Intention could be predicted by ML.
Comment: Theoretical implications should be discussed in light of previous findings in the tourism literature. While some more effort should be made to propose actionable managerial implications.
Response: the study focus includes the usage of mobile payments in domestic tourism and therefore the relevant literature cover both the aspects of the study i.e. tourism and mobile payments. A close analysis of the theoretical implications reveals that some of the implications are discussed in the tourism and mobile payment literature. For example, please see 49.
About the managerial implications, important development relevant to mobile payments such as GDPR, PSD2 have already been discussed in the manuscript. Nonetheless, implications concerning the attitude and behavior of the less privileged as well as the growing digital divide in developing countries in understanding and using ubiquitous technologies including mobile have been included in the revised manuscript.
Reviewer 2 Report
The article is interesting, but some issues require more precision:
1. What is the characteristics of the research place (Kazakhstan)? For what category does this country fall into, and on base of what criteria? What other countries are similar to Kazhstan in terms of these criteria?
2. Certain passages require bibliography to be completed (eg lines 82-93 or 165-167).
3. Lines 225-233 are a repeat of lines 190-198.
4. Incorrect numbering of the section title (line 305).
5. The authors should more strongly justify and explain the mechanism of purposeful sample selection in relation to the described issue. In this context, it should also be explained how to understand the p-value level mentioned in the text if we do not know the size of the entire population? Therefore, it is also worth asking the question of what the actual size of the population exhibiting the behavior under study may be?
6. The authors used a questionnaire based on the Likert scale. The Likert scale requires the computation of the Spearman correlation instead of the Pearson correlation. While Pearson's correlation is calculated not for questions but for key constructs, there is no precise explanation of how the questions turned into key constructs.
Author Response
Comment: The article is interesting, but some issues require more precision:
Response: We thank you for allowing us to revise and resubmit the manuscript. Point-by-point responses to your comments are discussed below.
Comment: What is the characteristics of the research place (Kazakhstan)? For what category does this country fall into, and on base of what criteria? What other countries are similar to Kazhstan in terms of these criteria?
Response: Kazakhstan and the domestic travellers and tourists therein were selected as the context of this study for four major reasons. Firstly, Kazakhstan is a highly collectivistic society [17] as it has a strong family system characterised by trust, harmony, and close ties between the family members and thus a strong familial influence. Social events and gatherings are also regularly arranged, and local travels are frequently undertaken. Secondly, the international travel restrictions that have been put in place globally due to the COVID-19 pandemic have promoted and increased the volume of domestic travel and tourism activities during holidays and on weekends in several countries [2,18], including Kazakhstan. Thirdly, such domestic travellers and tourists use mobile payment services frequently [19] as they consider these safer and more convenient than cash transactions while travelling with their family and friends to their dream destinations domestically. Lastly, the phenomenon of the use of mobile payment applications by domestic travellers and tourists coincides with the efforts of the government and other organisations to motivate the citizens to abandon or minimise cash transactions and to instead adopt remote or digital payment systems, including mobile payment systems.
Comment: Certain passages require bibliography to be completed (eg lines 82-93 or 165-167).
Response: We have rephrased the paragraph and added the necessary citations as mentioned below:
- Arbulú, I., Razumova, M., Rey-Maquieira, J., & Sastre, F. (2021). Can domestic tourism relieve the COVID-19 tourist industry crisis? The case of Spain. Journal of Destination Marketing & Management, 20, 100568.
- Law, R., Sun, S., Schuckert, M., & Buhalis, D. (2018). An exploratory study of the dependence on mobile payment among Chinese travelers. In Information and Communication Technologies in Tourism 2018 (pp. 336-348). Springer, Cham.
- Nunkoo, R., Daronkola, H. K., & Gholipour, H. F. (2021). Does domestic tourism influence COVID-19 cases and deaths?. Current Issues in Tourism, 1-14.
Comment: Lines 225-233 are a repeat of lines 190-198.
Response: Thank you for identifying this pitfall. We have removed the paragraph from section 3.1 to avoid repetition.
Comment: Incorrect numbering of the section title (line 305).
Response: We have revisited the whole document and rectified the mistakes/omissions in the section and sub-section serial numbers.
Comment: The authors should more strongly justify and explain the mechanism of purposeful sample selection in relation to the described issue.
Response: We apologize for this omission. The details concerning the survey data collection including the timelines and sampling technique have been added and explained in sub-section 4.1, which is now renamed as “4.1 Empirical Survey, Sampling, and Data Collection”.
Comment: It should also be explained how to understand the p-value level mentioned in the text if we do not know the size of the entire population? Therefore, it is also worth asking the question of what the actual size of the population exhibiting the behavior under study maybe?
Response: We have attempted to address this difficult issue. The population of interest is 19m (2020 figure), we did not intend to take a random sample but instead chose a purposeful sampling of informed respondents who would be able to provide data that could be validly analysed.
Comment: The authors used a questionnaire based on the Likert scale. The Likert scale requires the computation of the Spearman correlation instead of the Pearson correlation. While Pearson's correlation is calculated not for questions but for key constructs, there is no precise explanation of how the questions turned into key constructs.
Response: We have now clarified in Section 4.3 that the computations using Python Pandas library were for Pearson Correlation Coefficients. Key constructs were defined in the theoretic explanations in Section 2 and operationalised in Section 3.
Reviewer 3 Report
This article does not meet the level of the journal
2. The abstract part needs to add research gaps
3. The preface does not fully discuss the past Mobile Payment Systems and does not explain the research contribution
4. The Social Cognitive and Trust Theory research cannot support the research hypothesis
5. Discussion and Conclusion is incomplete
Author Response
Comment: This article does not meet the level of the journal
Response: We beg to differ.
Comment: The abstract part needs to add research gaps
Response: This has been done. Thank you.
Comment: The preface does not fully discuss the past Mobile Payment Systems and does not explain the research contribution
Response: Abstract and Introduction have been revised.
Comment: The Social Cognitive and Trust Theory research cannot support the research hypothesis
Response: We have since elaborated this aspect in Section 3.
Comment: Discussion and Conclusion is incomplete
Response: As the CRISP methodology was exploratory, we had not given too many details in the previous version. Nonetheless, considering your comments, Section 5 has now been expanded.
Round 2
Reviewer 1 Report
The authors addressed my comments. Well done!
Reviewer 3 Report
No commet.